# OpenReview forum: "RAIN: Your Language Models Can Align Themselves without Finetuning"
_ICLR.cc/2024/Conference — ICLR 2024 poster_

### Official Review · Reviewer_drDq · 2023-10-30

**Soundness:** 3 good
**Presentation:** 2 fair
**Contribution:** 2 fair
**Rating:** 5
**Confidence:** 4

**Summary:**

**Summary**

The paper introduces an inference method called RAIN, which allows pre-trained language models to align themselves without the need for finetuning or extra alignment data. This method integrates self-evaluation and rewind mechanisms to produce responses consistent with human preferences via self-boosting. The authors demonstrate the effectiveness of RAIN in improving the harmlessness and truthfulness rates of pre-trained language models through experiments on several benchmark datasets.

**Strengths:**

**Pros**

1. No need for extra alignment data: RAIN does not require any additional data for model alignment, which can save time and resources. This is because RAIN uses self-evaluation and rewind mechanisms to align the model, which allows it to learn from its own mistakes and improve its performance without the need for external supervision.

2. Improves model safety: RAIN can improve the harmlessness and truthfulness rates of pre-trained language models, which can make them safer to use in real-world applications. This is because RAIN allows the model to produce responses that are consistent with human preferences, which can reduce the risk of the model generating harmful or misleading outputs.

**Weaknesses:**

**Cons**

1. Limited to fixed pre-trained models: RAIN is designed to align fixed pre-trained language models and cannot be used to align models that are being trained or fine-tuned. This means that RAIN may not be suitable for applications that require continuous model alignment or adaptation.

2. Baselines: The paper does not provide a detailed comparison of RAIN with other methods that align pre-trained language models using RLHF or instruction tuning.

3. Limited evaluation on real-world scenarios: The experiments in the paper are conducted on benchmark datasets and may not fully capture the complexity and diversity of real-world scenarios. This means that the effectiveness of RAIN in real-world applications may be different from what is reported in the paper.

**Questions:**

**Questions for the authors**

How does RAIN compare to other methods that align pre-trained language models using reinforcement learning or instruction tuning?

What are the potential limitations or drawbacks of RAIN, and how can they be addressed in future work?

How scalable is RAIN to larger models or datasets, and what are the computational requirements for using RAIN on such datasets?

---

> ### Author Response · Authors · 2023-11-18
>
> Your thorough and insightful review has provided essential guidance for us. We are thankful for the opportunity to clarify and enhance our work based on your suggestions. Below, we address each of your queries and provide clarifications where needed, especially concerning the weaknesses you've identified.
>
> > Q1. Baselines: The paper does not provide a detailed comparison of RAIN with other methods that align pre-trained language models using RLHF or instruction tuning.
>
> Indeed, we have conducted such comparative experiments. As detailed in Figure 4 on page 8 of our paper, our comparison includes RLHF and RLAIF, both of which involve SFT (akin to instruction tuning) and RL phases. The results from these comparisons show that while RLHF and RLAIF require additional data, RAIN demonstrates improved performance.
>
> Furthermore, we have tested RAIN on models that were aligned using various methods. These experiments are presented in Table 1 on page 7, and in Table 2 on page 8. The results from these tables indicate that RAIN is capable of further enhancing the safety of models that have already been aligned using existing methods.
>
> > Q2. Limited to fixed pre-trained models: RAIN is designed to align fixed pre-trained language models and cannot be used to align models that are being trained or fine-tuned.
>
> Contrary to the impression that RAIN is restricted to fixed pre-trained models, it is indeed designed to work with any auto-regressive language model. Vicuna is a series of models fine-tuned from the LLaMA series. Our experiments demonstrate (as shown in Figure 1 on page 2, Figure 3 on page 7, and Table 1 on page 7) that RAIN is effective on both LLaMA and Vicuna. Similarly, LLaMA-2-chat is a series of models fine-tuned from the LLaMA-2 series. Our findings, as illustrated in Figure 1 on page 2, Figure 3 on page 7, and Table 2 on page 8, confirm the effectiveness of RAIN on both LLaMA-2 and LLaMA-2-chat. This underscores the fact that RAIN's effectiveness is not contingent on the fine-tuning process. We argue that inference with models that are actively being trained or fine-tuned inherently involves using a particular checkpoint, which is conceptually similar to using fine-tuned models like Vicuna (fine-tuned from LLaMA). Thus, RAIN's methodology is equally applicable and effective in these scenarios as well.
>
> > Q3. Limited evaluation on real-world scenarios: The experiments in the paper are conducted on benchmark datasets and may not fully capture the complexity and diversity of real-world scenarios.
>
> Thank you for your feedback. We have conducted human evaluations, which closely resemble assessments in real-world scenarios. The results, presented in Table 8 on page 9 of our paper, clearly show that RAIN also performs effectively under human evaluation. This approach not only validates the efficacy of RAIN in a controlled experimental setup but also demonstrates its practical applicability in real-world contexts.
>
> > Q4. What are the potential limitations or drawbacks of RAIN, and how can they be addressed in future work?
>
> In the second paragraph of Section 5, "CONCLUSION", we discuss the limitations of RAIN and propose avenues for future work. We acknowledge that compared to standard auto-regressive inference, RAIN demands a longer yet acceptable inference time. One potential solution to this issue, as we discuss, is to use RAIN to generate data and subsequently fine-tune the model with this data. This approach aims to offset the additional inference overhead by incorporating a fine-tuning phase.
>
> > Q5. How scalable is RAIN to larger models or datasets?
>
> One of the key advantages of RAIN is that it does not require additional memory overhead, which significantly aids its scalability to larger models. To demonstrate this, we have conducted extensive experiments with models as large as 70B parameters, which is the upper limit in the LLaMA series available to us. The results presented in Figure 1 on page 2 and Figure 3 on page 7, along with Table 1 on page 7 in our paper, clearly illustrate that the effectiveness of RAIN tends to increase with larger models. Furthermore, Table 7 on page 9 demonstrates that the relative additional time overhead required by RAIN diminishes as the size of the model increases. Additionally, RAIN is a universal, data-agnostic, and training-data-free algorithm, making it inherently scalable to large test datasets.

---

> > ### Comment · Reviewer_drDq · 2023-11-22
> >
> > Hi! I appreciate the authors' response and it does clear some of my confusion.
> >
> > However, some of my concerns remain.
> >
> > **Limited evaluation:** The authors show that they've conducted human evaluation, but I'm concerned about the task itself. Harmfulness/Helpfulness is one scenario. It would be much more interesting to investigate whether the method remains effective in general instruction-following tasks.
> >
> > **Computation burden:** The method has a high computational cost. This will make it challenging for the method to be deployed in practice.
> >
> >
> > I tend to keep my score unchanged.

---

> > > ### Author Response · Authors · 2023-11-22
> > >
> > > Thank you for your comments.
> > >
> > > > Q1. Limited evaluation: The authors show that they've conducted human evaluation, but I'm concerned about the task itself. Harmfulness/Helpfulness is one scenario. It would be much more interesting to investigate whether the method remains effective in general instruction-following tasks.
> > >
> > > Not only have we conducted experiments on Harmfulness/Helpfulness tasks, but we have also expanded our research scope in the experimental section of our original manuscript. This includes testing for adversarial robustness, truthful generation, and controlled sentiment generation, covering a broader range of general instruction-following tasks.

---

### Official Review · Reviewer_5Znu · 2023-10-30

**Soundness:** 2 fair
**Presentation:** 2 fair
**Contribution:** 2 fair
**Rating:** 6
**Confidence:** 4

**Summary:**

The paper proposes a new inference method called RAIN (Rewindable Auto-regressive INference) that allows large language models (LLMs) to align themselves with human preferences without requiring any finetuning or additional data.

- RAIN integrates self-evaluation and rewind mechanisms into the inference process. This allows the LLM to evaluate its own generated text, then "rewind" and modify the generation if needed to produce safer responses.
- Experiments on tasks like harm-free generation and truthful QA show RAIN improves alignment of fixed LLMs like LLaMA without finetuning.
- Advantages include no need for alignment data, compatibility with frozen LLM.

Overall, the paper demonstrates the feasibility of aligning fixed LLMs without any external data or training. RAIN provides an inference-time solution to improve safety and alignment that is user-friendly, resource-efficient, and broadly applicable.

**Strengths:**

The paper studies the important topic of aligning LLMs. The paper takes a specific angle of designing inference-time techniques to reduce the changes that models output harmful text.

**Weaknesses:**

- The proposed inference-time method has high computational cost. This will make it challenging for the method to be deployed in practice. (On the other hand, one can perform fine-tuning on generated samples so that the sampling approach need not be performed at inference time; the authors didn't not explore this approach.)
- The proposed method is complicated.
- The authors use an umbrella "harmless" notation (for which the self-evaluation score is based on), whereas in practice whether something counts as harm can be contextual.

**Questions:**

NA

I read the author response, and I am keeping my original score.

---

> ### Author Response · Authors · 2023-11-18
>
> Thank you for your comprehensive review and the valuable insights offered. Your feedback is instrumental in helping us improve our manuscript. In the following, we have made an effort to clarify the aspects of our work that you found lacking.
>
> > Q1. The proposed inference-time method has high computational cost. This will make it challenging for the method to be deployed in practice. (On the other hand, one can perform fine-tuning on generated samples so that the sampling approach need not be performed at inference time; the authors didn't not explore this approach.)
>
> In the conclusion section, we discuss the limitations of RAIN: Compared to standard auto-regressive inference, RAIN requires a longer, yet acceptable, inference time. A potential solution to this issue could involve using RAIN to generate data and then fine-tuning models with this data. However, fine-tuning LLMs is resource-intensive and contradicts the aim of our paper, which is to "investigate the feasibility of enabling (self-)alignment in fixed LLMs without fine-tuning." Therefore, we have deferred this approach to future work.
>
> > Q2. The proposed method is complicated.
>
> Intuitively, RAIN conducts Monte-Carlo search on the tree consisting of token sets and dynamically reduces the weight of harmful token sets, with backward rewind and forward generation steps until the output content is self-evaluated as harmless. We also modify the Monte-Carlo tree search to adapt to the LLM alignment tasks by introducing the token similarity check. The potentially "complicated" design is to improve the search efficiency.
>
> > Q3. The authors use an umbrella "harmless" notation (for which the self-evaluation score is based on), whereas in practice whether something counts as harm can be contextual.
>
> We chose the umbrella term "harmless" to maximize the generalizability of our method. Our experiments demonstrate that utilizing a generalized "harmless" notation for self-evaluation allows the model to identify different types of harmful content effectively, thus enhancing RAIN’s universal applicability in assessing harmlessness. We also recognize that adopting more specific terms, such as "privacy," could potentially yield better results in assessing specific types of harm. We appreciate your comment and agree that harm must be considered in context. In fact, the self-evaluation in RAIN and its performance evaluation (whether using GPT-4 or human evaluation) take context into account. The <Generated Text> in the prompt examples on page 5 actually include context. Replacing <Generated Text> with <Contextual Generated Text> or <Query and Generated Text> would provide greater clarity. Your suggestion is highly valuable.

---

### Official Review · Reviewer_RrQ5 · 2023-10-31

**Soundness:** 3 good
**Presentation:** 2 fair
**Contribution:** 2 fair
**Rating:** 6
**Confidence:** 3

**Summary:**

This paper proposes a new inference-time method to align LLMs' output to human preference via nested iterations. In the outer loop, the algorithm will run the subsequence generation process ("inner loop"), concatenate subsequences to the final outputs, and keep checking whether the current generated sequence already contains the end-of-sequence token. In the inner loop, the proposed algorithm will generate sub-sequences via an iterative search process: 1) doing lookahead planning in a tree-search paradigm ("forward") and 2) self-evaluating the newly added nodes and updating values for nodes in the current search tree through the node ("backward"). When the termination condition is reached (e.g., the maximum number of iterations, or the value of the most-visited child node from the root is larger than a certain threshold), this iterative search process ends. The output of this search process is the token sequence associated with the tree path from the most-visited child node from the root. The experiment result demonstrates the effectiveness of the proposed method over harm-free generation, adversarial harm-free generation, truthful generation, and controlled sentiment generations via both automatic generation and human evaluation, though induces significant computational overhead.

**Strengths:**

1. This paper provides a model-agnostic inference-time algorithm that can save the training cost of doing alignment, and it also provides evidence that incorporating self-evaluation in search-based decoding can help improve the model alignment.

2. The authors have done extensive experiments on various tasks and models to verify the effectiveness of the proposed method via both automatic evaluation and human evaluation, which makes this paper more solid.

**Weaknesses:**

1. **Limited Applications**: The proposed nested iterative search algorithm significantly increases the computational overhead at inference time (>3.7x on 70B models, and >4.3x on 30B models), making it intractable for practical purposes as the running time for these large models are already very large. In the meanwhile, the tree-search operation makes it extremely hard to perform parallelized optimization and can induce significant I/O throughputs as many operations in this proposed algorithm may not be able to easily run on GPU, so batch inference will be much harder to implement, further increasing the real latency on the user side. Considering this significant computational overhead, it is not clear whether this performance gain reported in this paper is worth it -- maybe practitioners should instead sample many more outputs via some diversity-enforced sampling method [1] (parallelizable and has been largely optimized via various open-source libraries such as VLLM [2]), and apply a well-tuned reward model to pick up the best outputs (note that this best-of-K/rejection sampling method is not even listed as baseline in this work).

2. **Partially-Justified Motivation, and perhaps Marginal Contributions**: The author argues that "Beyond performance, the primary goal of this paper is to investigate the feasibility of enabling self-alignment in fixed LLMs without fine-tuning or RL" (in Introduction). However, it is not clear the necessity of applying the proposed iterative tree-search algorithm to verify this feasibility. As the authors recognize in the paper, there are already several tree-search-style inference-time algorithms in adjusting LLM outputs (the author mentioned [3] in the paper, and I think [4] is also related), and the self-evaluation does not seem like a new contribution in decoding content planning to improve both LLM reasoning and alignment (e.g., [1,5,6]) without learning. Though these works may not run on the specific task set as the author, they do have the potential to be strong competitors and some of them have already achieved the verification that a "learning-free algorithm can improve alignment via self-evaluation".

3. **Weak Baselines**: As explained in the above points, best-of-K/reject sampling, and existing tree-search algorithm (without iterations and back-tracking. Introducing self-evaluation to them does not seem too hard -- correct me if I am wrong.) have full potential to be the strong competitors to the proposed method (some of them may have already been deployed somehow by companies), given sufficient computational budgets (perhaps 3-4x more than normal inference time, to make a fair comparison to the proposed algorithm). None of them is made as a baseline in this paper and the author mainly compares with vanilla auto-regressive generation, which is relatively weak. Note, this is not to say the author needs to achieve state-of-the-art when comparing with these stronger baselines -- this is to help users understand the necessity of deploying the proposed algorithm and choose based on their needs.

4. **Unclear Experiment Details**: What are the maximum number of iterations $T$ and the minimum number of iterations $T_m$ used in the experiment? How many tokens are generated for each node? Moreover, how does the author decide each hyperparameter for each task? I checked both the main text and appendix and I did not find how the author set these parameters and based on what observations. These parameters (especially $T$ and $T_m$)  can be very important for both reproduction purposes and for users to see a trade-off between efficiency and performance if they want to deploy the proposed method.

5. **Hard-to-Read Formulation**: It is very hard to read the formulation presented in Section 3. For example, should not the `node` use different notations from the actual `token`? As each node contains more `attributes` (visit counts, embeddings) than a token? In many places, the authors use the same symbol to represent tokens and nodes. Also, $X$ and $Y$ sometimes seem interchangeable (e.g., $Y_{1:i-1}$ and $X_{1:i-1}$ as the "preceding context"). The usage of the subscript is not careful -- for example, why $S(Y)$ represents the score used to update $Y^*_{a:b}$ without having $a,b$ as input to $S()$?

References:

[1] Bai, Yuntao, et al. "Constitutional ai: Harmlessness from ai feedback." arXiv preprint arXiv:2212.08073 (2022).

[2] Kwon, Woosuk, et al. "Efficient Memory Management for Large Language Model Serving with Paged Attention." SOSP 2023.

[3] Yao, Shunyu, et al. "Tree of thoughts: Deliberate problem solving with large language models." NeurIPS 2023.

[4] Zhao, Hongyu, et al. "Explicit Planning Helps Language Models in Logical Reasoning." EMNLP 2023.

[5] Shinn, Noah, et al. "Reflexion: Language Agents with Verbal Reinforcement Learning." NeurIPS 2023.

[6] Ganguli, Deep, et al. "The capacity for moral self-correction in large language models." arXiv preprint arXiv:2302.07459 (2023).

**Questions:**

Most of my questions can be found in my "Weakness" section, I just list some here:
1. Why other inference-time algorithms are not listed as baselines?
2. What are the detailed efficiency numbers (in detailed wall clock time instead of ratio) for your proposed method?
3. What are some missing hyperparameters?
4. How do you decide them for each task?
5. Can you review the formulation in Section 3 and update it to be more clear?
6. Is there any possibility of achieving a better efficiency-performance trade-off using your method? For example, reducing the number of iterations?

---

> ### Author Response · Authors · 2023-11-18
>
> We greatly appreciate your insightful comments and constructive criticism. They are invaluable in guiding us to refine our manuscript. Here are our detailed responses to your points and further explanations to address the concerns raised.
>
> > Q1. Best-of-K/rejection sampling method is not listed as baseline in this work.
>
> In fact, we have conducted a comparison with the method you mentioned. This is detailed in Table 5 on page 9 of our paper, where the  best-of-K method is referred to as "Vanilla (Repeated)." Best-of-K is a specific case of rejection sampling, and if the evaluation is effective, best-of-K should not perform worse than rejection sampling. The experimental results demonstrate that even with 500 trials (best-of-500), for the 7B model, best-of-K barely improves the model’s safety. With the 30B model, RAIN shows a significantly enhanced performance. This clearly indicates the superior efficacy of RAIN compared to the best-of-K/rejection sampling approach.
>
> > Q2. Even though other related works have not been applied to the specific task set used by the author, they still have the potential to be strong competitors.
>
> For [1] (RLAIF), we have included comparative results in Figure 4 on page 8. The method described in [6] is applied to models trained with RLHF, which diverges from our focus on non-finetuned models. To the best of our knowledge, for the methods in [3,4,5], there is no clear evidence suggesting their applicability in the domain of LLM safety. The method in [5] necessitates interaction with the environment to gather feedback, which, in our research context, would mean requiring real user feedback. This is impractical in the field of safety and makes a comparison with RAIN unfair. The problems addressed by RAIN and those tackled by the aforementioned studies are fundamentally different. RAIN's external form is consistent with that of auto-regressive generation, which makes it adaptable to a wide range of language generation tasks. The studies in [3,4] are primarily concerned with reasoning tasks, which inherently allow for explicit planning. For example, [3] investigates tasks like the 24-point game, which involves multi-step number combinations, while [4] focuses on logical reasoning, which can be broken down into multi-step selection and deduction processes. Such explicit planning structures are absent in tasks like harmless generation, leading to different search spaces. The search space in [3,4] revolves around actions like combining numbers, whereas RAIN's search space is broader, encompassing the entire language space.
>
> Despite various challenges, we made our utmost effort to adapt LEAP proposed in [4] for the task of harmless generation and subsequently conducted experiments on the HH dataset. Here are the experimental details and results. We replaced the actions in LEAP with the selection of token sets. LEAP uses $u=\log p_{\text {sel }}(\mathbf{s})+\alpha \Delta u$ and $\Delta u = \max_d \log p_{\text{ver}}\left(x_0 \mid y_d\right)$ to determine the score, where $p_{\text {sel}}$ is given by the selection model and $p_{\text{ver}}$ by a different verification model. As we did not train a verification model, we set $p_{\text {sel}}$ to be the probability obtained from querying LLMs and $p_{\text{ver}}$ as the normalized self-evaluation score, ensuring the meanings of both terms remain unchanged. The experiment's hyper-parameters were consistent with the original paper of [4], such as $B=5$, $\alpha=10$, etc. The results are as follows:
>
> |      Method       | Vanilla |  LEAP  | RAIN  |
> |:-----------------:|:-------:|:------:|:-----:|
> | harmlessness rate |  82.2%  | 89.1%  | 97.0% |
>
> Our findings indicate that while LEAP improves harmlessness over vanilla auto-regressive generation, RAIN significantly outperforms both methods.
>
> > Q3. Unclear experiment details. What are the maximum number of iterations $T$ and the minimum number of iterations $T_m$ used in the experiment? How many tokens are generated for each node?
>
> Thank you for pointing it out. In RAIN, hyper-parameters $T_m$ and $V$ control the early exit strategy, balancing time cost against performance. These parameters are tuned based on the accuracy of self-evaluation: the more precise the self-evaluation, the lower we can afford to set $T_m$ and $V$. For all tasks except truthfulness, we set $T_m=2$ and $V=0.7$, whereas for truthfulness, due to its increased complexity, we used $T_m=16$ and $V=0.8$. Across all tasks, the upper limit of the inner loop iterations, represented by $T$, was fixed at 50. When generating responses, tokens positioned earlier in the sequence have a greater influence on the entire token sequence. Therefore, we set a varying number of tokens for nodes at different positions, specifically in the sequence of 1, 2, 4, 10 (with all subsequent nodes containing 10 tokens each).

---

> > ### Author Response · Authors · 2023-11-18
> > **Response continued**
> >
> > > Q4. Should not the node use different notations from the actual token?
> >
> > Thank you for your insightful suggestion. In our paper, nodes encompass more than just the token set, including four additional attributes: embedding, probability, visit count, and value. Acknowledging this complexity, it would indeed be clearer to differentiate between "node" and "token set" with distinct notations, enhancing the clarity and precision in our method section.
> >
> > > Q5. $X$ and $Y$ sometimes seem interchangeable.
> >
> > As mentioned in the notation section, uppercase letters like $X$ and $Y$ represent sequences of tokens (token set). We use $X$ and $Y$ simultaneously in some places to denote two distinct token sets, such as those corresponding to child and parent nodes.
> >
> > > Q6. Why does $S(Y)$ represent the score used to update $Y_{a:b}^{*}$ without having $a,b$ as input to $S()$?
> >
> > In our method, the formula for updating is as follows:
> > \begin{equation*}
> > v(Y_{a:b}; Y_{1:a-1})=\frac{1}{\left| {X: Y_{1:b}=\operatorname{prefix}(X)} \right|}\sum_{{X: Y_{1:b}=\operatorname{prefix}(X)}} s(X).
> > \end{equation*}
> > This formula indicates that we update $Y_{a:b}$ using $s(X)$. While $a$ and $b$ are not explicit inputs to $S()$, they are implicitly linked through the condition $X: Y_{1:b}=\operatorname{prefix}(X)$. Here, $Y_{1:b}$ serves as a prefix of $X$, and $a$ can range from 1 to $(b-1)$.
> > The reason for this update is that token sets which are prefixes of $X$ contribute to $s(X)$, and $s(X)$ should update all token sets that contribute to it. We also provide an example in the paper: the value for "Robbing is" should be the average score of "Robbing is illegal" and "Robbing is a serious offense."
> >
> > > Q7. What are the detailed efficiency numbers (in detailed wall clock time instead of ratio) for your proposed method?
> >
> > The majority of the computational overhead for RAIN arises from LLMs inference. Consequently, the absolute runtime is closely tied to the experimental setup, particularly the GPU configuration. In our experiments with the HH dataset, the LLaMA 30B model using RAIN averaged 67.8 seconds per data point, and the vanilla auto-regressive averaged 15.6 seconds.
> >
> > > Q8. Is there any possibility of achieving a better efficiency-performance trade-off using your method? For example, reducing the number of iterations?
> >
> > To achieve a better efficiency-performance trade-off using our method, it is essential to either enhance the performance without increasing the computational burden or reduce the computational overhead without adversely affecting performance. A potential solution for this is to employ a more precise reward model for evaluation. By improving the accuracy of self-evaluation, this approach aims to achieve enhanced performance without escalating the computational costs. However, it is important to note that this strategy may require extra data to develop and refine the reward model. While reducing the number of iterations might lower computational costs, it is important to consider that it could also impact the method's overall effectiveness.

---

> ### Comment · Reviewer_RrQ5 · 2023-11-20
> **Responses for Rebuttal (part.1)**
>
> Thanks authors for clarifications. I have further questions after reading your response:
>
> 1. Regarding best-of-K sampling as a baseline:
>
>     I see that I misunderstood your setup of making "Vanilla (Repeated)" as best-of-K sampling. I strongly suggest you have a section of `baseline` in your formal version explaining all baselines you plan to use for clarification, explaining what baselines you are implementing, how you set up hyperparameters for them, and how you adapt them to your experimental setting if necessary, etc. Currently, there is no clear explanation for it to be regarded as a best-of-K sampling baseline. However, it adds more confusion as to why this can be a best-of-k sampling implementation. First, there is no explanation of your sampling parameters (the temperature, the `top_p` and `top_k`) -- this is very important as we need to select from a **diverse** set of generations. Second, why do you think using LLM itself as an evaluator is a valid best-of-k sampling (perhaps you can mention some work to support your decision, but I do not see any such support in the current version)? Typically, the best-of-K sampling method is implemented by using another "preference model" to pick up the best one as the final answer ([1], Section 2.3). If you use the same model to select its outputs, there might be strong biases put on the maximum likelihood one, making the output not significantly different from "Vanilla" (as evidenced by LLaMA-7B results in Table 5). Considering there are many open-source preference models already released online, I think a valid best-of-K sampling baseline should be at least using one of these models (e.g., OpenAssistant Reward Model: https://huggingface.co/OpenAssistant/reward-model-deberta-v3-large).
>
>     In short, if you want to justify Vanilla (Repeated) as a valid best-of-K baseline, more details and perhaps some more experiments need to be conducted.
>
> 2. Regarding Comparison to Previous Works
>
>     Thanks for the detailed explanation. I again strongly suggest you add a more detailed section on `baselines` and include your discussions here. I appreciate your efforts in replicating some of my mentioned previous works and it is great to see your proposed method can give such significant performance boosting.
>
> 3. Regarding Missing Experiment Details
>
>     Thanks for the explanations. Please add these parameter specifications in your paper as well as your justification on setting them.
>
> References:
>
> [1] Bai, Yuntao, et al. "Training a helpful and harmless assistant with reinforcement learning from human feedback." arXiv preprint arXiv:2204.05862 (2022).

---

> ### Comment · Reviewer_RrQ5 · 2023-11-20
> **Responses for Rebuttal (part.2)**
>
> Q4-Q6: Thanks for the explanation. Please re-read your current formulation and update it appropriately.
>
> Q7: I agree the detailed running time strongly depends on your experimental setup and can vary by different computational infrastructures underlying your experiments. Thanks for providing these efficiency numbers. I think it is a good illustration of the points I explained in "limited applications": the proposed decoding method is not easy to parallelize, thus the computational overhead seems to overwhelm the performance benefit it can bring. Also, it does not seem to be easy for practitioners to incorporate the proposed method in decoding time from an engineering perspective.
>
> Q8: Thanks for proposing a potential solution to achieve a better efficiency-performance trade-off, although it perhaps makes your method closer to best-of-K sampling/rejection sampling as proposed in [1]. As I explained in my responses (part.1), there are already many preference model/reward models available online. You do not necessarily need to collect your data to train a new reward model. Also, there are already many human preference data online and it seems straightforward to train your own reward model if you are not satisfied with the open-sourced reward model. With that said, I want to clarify that this question is more about a discussion for future extension, and does not constitute a criticism (that is why I only listed it in the "Questions" section).
>
> *Additional Notes to facilitate discussions*: Except for reducing the number of iterations to achieve better efficiency, I originally thought of not doing this tree search for a proportion of tokens generated and using vanilla auto-regressive generation instead. Using the human utterance generation process as an analogy, we do not need to consider that carefully for every token when the context is clear (you also observe that not all tasks require the same (large) number of iterations, and $T=0$ is just a special case). Not sure whether the author may find it a useful and interesting idea to explore. Just write it in case the author may find it useful.

---

> ### Comment · Reviewer_RrQ5 · 2023-11-20
> **General Comments after Reading the Rebuttal**
>
> I appreciate the detailed responses from the author and it does clear many of my confusions. However, the limited application range (due to computational overhead), unclear explanation of baseline settings, etc. are still my concerns. Based on these considerations, I would raise my score to 6.

---

> > ### Author Response · Authors · 2023-11-22
> >
> > We are sincerely grateful for the recognition and the improved evaluation of our work. Your constructive feedback and insightful questions have been instrumental in enhancing our manuscript. Below, we provide comprehensive responses to your queries and further clarifications to address your concerns.
> >
> > > Q1. Regarding best-of-K sampling as a baseline: I strongly suggest you have a section of baseline in your formal version explaining all baselines you plan to use for clarification. Currently, there is no clear explanation for it to be regarded as a best-of-K sampling baseline. First, there is no explanation of your sampling parameters (the temperature, the top_p and top_k) -- this is very important as we need to select from a diverse set of generations. Second, why do you think using LLM itself as an evaluator is a valid best-of-k sampling? I think a valid best-of-K sampling baseline should be at least using one of these models (e.g., OpenAssistant Reward Model: https://huggingface.co/OpenAssistant/reward-model-deberta-v3-large).
> >
> > We appreciate your valuable suggestion and will include a section of baselines in the final version of our manuscript for clearer understanding. In our "Vanilla (Repeated)" approach, we utilized direct sampling from the LLM's original distribution, i.e., setting the temperature parameter to 1, without employing additional sampling strategies such as top_k and top_p. We also conducted additional experiments with different sampling parameters, including those involving a preference model. The results of these experiments are presented in this response.
> >
> > Using LLM itself as an evaluator may not be the optimal choice for best-of-k sampling. However, under the premise of ensuring comparability and fairness in our experimental setup, we consider it a valid approach. The primary aim of our research is to investigate the feasibility of enabling (self-)alignment in fixed LLMs, rather than solely focusing on performance. Consequently, we restricted RAIN from accessing preference models trained with external knowledge. To ensure a fair comparison, "Vanilla (Repeated)" was evaluated using the same self-evaluation method as RAIN, rather than an external preference model.
> >
> > Additionally, we have conducted experiments using the preference model you mentioned (OpenAssistant Reward Model: https://huggingface.co/OpenAssistant/reward-model-deberta-v3-large). In these experiments, we scored the generations with the preference model and selected the highest-scoring outputs as the final results. The findings from these experiments, conducted using the LLaMA 30B model on the HH dataset, are as follows:
> >
> > |      Setting      | temperature=1, top_p=NA, top_k=NA | temperature=0.5, top_p=NA, top_k=NA | temperature=1, top_p=0.9, top_k=NA | temperature=1, top_p=NA, top_k=60 | RAIN |
> > |:-----------------:|:---------------------------------:|:-----------------------------------:|:---------------------------------:|:-------------------------------:|:----:|
> > |   self-evaluation |              88.1%               |                86.6%                |              87.6%               |              88.5%              | 97.0%|
> > | preference model  |              92.5%               |                91.1%                |              92.1%               |              92.1%              |  NA  |
> >
> > The experimental results show that while there are differences in outcomes due to various sampling strategies, these differences are not substantial. The experimental results also indicate that employing a preference model yields better outcomes compared to self-evaluation. This disparity could be attributed to the model bias inherent in self-evaluation. Notably, despite this, RAIN, which utilizes self-evaluation, still outperforms the best-of-K method that uses a preference model.
> >
> > > Q2. Thanks for proposing a potential solution to achieve a better efficiency-performance trade-off, although it perhaps makes your method closer to best-of-K sampling/rejection sampling as proposed in [1]. As I explained in my responses (part.1), there are already many preference models/reward models available online.
> >
> > Thank you for your insightful comments. If the primary goal were solely to enhance performance or achieve a better efficiency-performance trade-off, utilizing preference models/reward models would indeed be highly effective and likely yield superior results. Implementing such an approach would simply involve substituting self-evaluation with scoring by preference models/reward models. It is worth noting that, in addition to aiming for performance improvement, our primary objective is to "investigate the feasibility of enabling (self-)alignment in fixed LLMs without finetuning". Hence, we did not conduct experiments involving these models, as they would be more suitable for discussion in future work.

---

> > > ### Author Response · Authors · 2023-11-22
> > > **Response continued**
> > >
> > > > Q3. Additional Notes to facilitate discussions: Except for reducing the number of iterations to achieve better efficiency, I originally thought of not doing this tree search for a proportion of tokens generated and using vanilla auto-regressive generation instead. Using the human utterance generation process as an analogy, we do not need to consider that carefully for every token when the context is clear (you also observe that not all tasks require the same (large) number of iterations, and $T=0$ is just a special case). Not sure whether the author may find it a useful and interesting idea to explore. Just write it in case the author may find it useful.
> > >
> > > Thank you for your idea. As you mentioned, for a given sentence, while some tokens are very clear, others require careful consideration. Allocating computational resources to tokens that need this careful consideration can indeed achieve better efficiency. RAIN makes use of this aspect to some extent with its early exit mechanism. This mechanism, though, does not specifically identify which tokens significantly influence the final output. Identifying non-critical tokens is challenging without finetuning or using additional data. If there were no restrictions, training a head or model to identify non-critical tokens could be a potential solution. Thank you for your suggestion; it is both useful and interesting.

---

> ### Comment · Reviewer_RrQ5 · 2023-11-22
> **Thanks for the follow-ups!**
>
> I really appreciate the author's timely follow-up experiments, explanations, and insightful discussions. Please make sure to add them to the upcoming version. My main concerns now are only on computational costs (as other reviewers also point out) and moreover, I am not fully convinced yet that we need to develop such complicated methods to verify that the model can do (self)-alignment -- the question that confused me most is that, if we really need a complicated method design to make the model to do self-alignment, is it really "self"-alignment? I am worried maybe we are risking treating some other emergent capabilities as "self"-alignment.
>
> Also here is another conceptual question (open to discussions with the author, other reviewers, chairs, and public reviewers are welcome): Let's ignore the methodology part for a while, if we can observe the model can do self-alignment, then why do we need to do "alignment"? By definition, self-alignment already means the model should be able to align itself really well, and should not need help doing alignment. This is **not a criticism**, instead, it is more of a research question that is hanging in my mind, and curious to hear people's opinions.
>
> I realized that the discussion phase is over and considering my timeline, I am no longer for further discussions. Thanks to the author for the hard work on discussing with me and providing follow-ups. I tend to keep my scores unchanged.

---

### Official Review · Reviewer_WBkA · 2023-11-01

**Soundness:** 3 good
**Presentation:** 3 good
**Contribution:** 4 excellent
**Rating:** 8
**Confidence:** 3

**Summary:**

This paper presents an intriguing and innovative method called Rewindable Auto-regressive INference (RAIN) to address the alignment problem of large language models (LLMs). Unlike previous fine-tuning-based approaches such as DPO, RLHF, and RAFT, which require additional data and optimization processes, the proposed method leverages LLMs' self-evaluation capacity and rewind decoding method to enable LLMs to align themselves with plausible preferences. It resembles human weighing patterns, involving evaluation, backtracking, and the selection of optimal generation paths. It is worth noting that while RAIN shares similar decision-making logic with Tree-of-Thought, the targeted problems and detailed self-evaluation mechanisms are distinct.

In summary, this paper makes the following contributions:

1. It introduces a novel self-alignment method for LLMs that does not rely on additional training resources, opening up an exciting research direction for addressing LLM alignment.

2. It demonstrates that LLMs are capable of performing self-alignment. Moreover, compared to existing approaches, it achieves remarkable alignment performance while requiring fewer resources.

3. It enhances the safety of mainstream open-sourced LLMs such as LLaMA-family, Vicuna, Alpaca, etc., without compromising the helpfulness of their outputs.

**Strengths:**

1. **Significance**: Aligning LLMs with correct values is an issue of utmost importance in the development of trustworthy and secure AI. Previous approaches require the use of additional models or SFT data to train LLMs on alignment preferences, which significantly increases the training cost. The proposed method enables LLMs to align themselves without external data, thereby reducing the development cost of LLMs even further.

2. **Originality**: This paper is the first to propose the use of LLMs for self-alignment, which not only represents a highly novel contribution but also establishes a new research paradigm for LLM alignment. I am looking forward to further work that focuses on reducing the inference cost of self-alignment.

3. **Quality**: The paper is written in a clear and easily understandable manner. The algorithm explanation is well-supported with illustrative examples, aiding in comprehension. Furthermore, the experimental setup covers mainstream open-sourced LLMs, and the comparison experiments include discussions on other alignment methods, showcasing the thoroughness of the work.

**Weaknesses:**

1. **Clarity**: There are some minor clarity issues in this paper:
   - Algorithm 1 is not referenced in the main text, although it is presented in the methodology section. It would be helpful to mention Algorithm 1 and provide a brief explanation or reference to it where relevant.
   - In the evaluation and searching process, there are several thresholds mentioned, such as $T_m$ and $V$. It would be beneficial to clarify whether these thresholds have common settings or if they are benchmark-specific.

**Questions:**

1. In the inner loop, could alternative LLMs be utilized for conducting evaluation? What are your thoughts on the potential benefits of employing multi-agent debating for self-evaluation?

2. RAIN employs the embedding of pre-trained Sentence-BERT for calculating cosine similarity. How do you perceive the possibility of replacing it with the embedding from the LLM itself?

---

> ### Author Response · Authors · 2023-11-18
>
> Thank you for your acknowledgement! Your thorough reading and insightful comments have been tremendously helpful to us. This response addresses all of your questions.
>
> > Q1. Algorithm 1 is not referenced in the main text, although it is presented in the methodology section. It would be helpful to mention Algorithm 1 and provide a brief explanation or reference to it where relevant.
>
> Thank you for highlighting the oversight regarding Algorithm 1. In the revised version of our paper, we will ensure that Algorithm 1 is clearly referenced. Furthermore, in the methodology section, we will add correspondences between the text and the pseudocode of the algorithm to enhance clarity and understanding.
>
> > Q2. In the evaluation and searching process, there are several thresholds mentioned, such as $T_m$ and $V$. It would be beneficial to clarify whether these thresholds have common settings or if they are benchmark-specific.
>
> Thank you for your insightful suggestion. In our methodology, we utilize thresholds $T_m$ and $V$ to effectively manage the early exit mechanism, aiming to balance time efficiency with performance. The calibration of these parameters depends heavily on the accuracy of the model's self-evaluation. Generally, the more accurate the self-evaluation, the lower we can set these thresholds.
>
> In our experiments, we identified the truthfulness task as particularly challenging for self-evaluation, requiring different threshold settings. Specifically, we used $T_m=2$ and $V=0.7$ for all tasks except truthfulness. For the truthfulness task, the parameters were adjusted to $T_m = 16$ and $V = 0.8$ to accommodate its unique challenges. The upper limit $T$ for the inner loop was consistently set at 50 across all tasks.
>
> Regarding response generation, earlier tokens in a sequence have a more pronounced influence on the overall arrangement. To address this, our node structure is designed to vary the token counts at different positions in the sequence. The structure follows a pattern of 1, 2, 4, and 10 tokens, with subsequent nodes containing 10 tokens each.
>
> > Q3. In the inner loop, could alternative LLMs be utilized for conducting evaluation?
>
> Using alternative LLMs for evaluation in the inner loop is an interesting idea. It could reduce biases, especially if the evaluation LLMs differ from the generation model. It is important to note, though, that if the models are similar, the extent of bias reduction might be limited. Additionally, this approach increases memory overhead due to the need to store both generation and evaluation models.
>
> > Q4.  What are your thoughts on the potential benefits of employing multi-agent debating for self-evaluation?
>
> The idea of employing multi-agent debating for self-evaluation is indeed promising. It has the potential to facilitate more accurate assessments, potentially leading to the generation of higher-quality responses. Multi-agent debating is computationally more intensive. This increased computational demand suggests that its most effective application might be in scenarios where immediate response times are not critical, allowing a greater focus on maximizing output quality.
>
> > Q5. RAIN employs the embedding of pre-trained Sentence-BERT for calculating cosine similarity. How do you perceive the possibility of replacing it with the embedding from the LLM itself?
>
> Thank you for your comment. This method has the advantage of not requiring an additional model or extra computation during feature extraction. It is noted that the significant size difference between Sentence-BERT and LLMs results in a minimal difference in computational cost. We conducted experiments under the settings of our ablation study for this aspect. In our experiments, we utilized the last token's feature from the last layer's hidden state of the LLM, as it encapsulates the context of the entire sentence, following the standard approach for sentence classification with LLMs. Our experimental results are as follows:
>
> | Embedding           | None | LLMs itself | Sentence-BERT |
> |:-------------------:|:----:|:-----------:|:-------------:|
> | Attack Success Rate↓| 25%  | 22%         | 19%           |
>
> The results show that while using LLMs for embedding extraction outperforms omitting embedding-related mechanisms (Dynamic Node Addition \& Similarity Update), it is less effective than utilizing Sentence-BERT. This is likely due to Sentence-BERT being specifically trained for sentence embeddings, thereby providing higher-quality embeddings compared to LLM features, which are primarily designed for next-token prediction.

---

> > ### Comment · Reviewer_WBkA · 2023-11-22
> > **Reviewer's Response**
> >
> > I would like to thank the authors for their responses and extend my appreciation to the other reviewers for discussing this paper. I still acknowledge the novel perspective introduced by the authors in this research direction.
> >
> > Regarding Q1 and Q2, I appreciate the authors' efforts in elucidating the search settings. I recommend incorporating these explanations into the subsequent version of the paper.
> >
> > For Q3-Q5, the authors have provided justifications and outlined the prospective enhancements to the methodology design. I am pleased to note that certain points raised during the review process may contribute to further strengthening the proposed methodology.
> >
> > While my overall rating remains unchanged, I concur with the observations made by other reviewers concerning *the need for improved clarification about experimental settings* in the paper. Additionally, I suggest that the authors include a *more comprehensive discussion regarding the practical applicability (such as computational cost)* of the proposed approach.
> >
> > I anticipate witnessing the evolution of this work in a future version.

---

> > > ### Author Response · Authors · 2023-11-22
> > >
> > > Thank you for your recognition. We agree with your suggestion and will include a detailed description of the search settings in the subsequent version of our manuscript. In line with the recommendations from the reviewers, we are committed to making improvements in the upcoming version, especially in terms of clarifying the experimental settings and detailing the computational costs.

---

### Official Review · Reviewer_hgqg · 2023-11-02

**Soundness:** 2 fair
**Presentation:** 2 fair
**Contribution:** 3 good
**Rating:** 5
**Confidence:** 3

**Summary:**

This paper is interested in *self-alignement*, i.e to align a model with human values without external supervision. Instead of proposing a method involving fine-tuning or any parameter update, it investigates a *rewind* mechanism during the inference phase, which, coupled with self-evaluation, allows the model to appraise generated text and only output something after having evaluated and eventually retraced its steps. The authors advertise three main advantages for their approach: being applicable to any model using auto-regressive inference, keeping the model frozen with no parameter update necessary, and not necessitating human supervision.

The paper presents related work on alignment with and without reinforcement learning, and backtracking in inference, then, the method, Rewindable Auto-regressive Inference (RAIN), which consists in an *inner* and an *outer* loops. Assuming a tree structure representing the next possible token sets,
- The inner loop picks a leaf set, trying to balance exploration and exploitation of the token set value, and eventually extends the tree if necessary. It computes scores using a *self-evaluating prompt* which is used to update values for the parents in the tree (and the rest of the tree, through similarity of embeddings of token sets). It then samples new token sets.
- The outer loop uses attributes (probabilities, values, and number of visits from the inner loops) to pick the next token set to be generated, similarly to a classical inference method.

The authors evaluate their approach with 4 tasks: Harm-free generation, robustness to adversarial harmful generation, truthful generation, and controlled generation. The result of the model's generation process on these tasks is then evaluated by GPT-3 and 4 models, and human evaluation is further carried out on Harm-free generation. The authors also analyse their model, looking at sample and time efficiency, the accuracy of self-evaluation in the inner loop, and performing an ablation study.

**Strengths:**

- This paper proposes an approach for self-alignment, requiring no human supervision, no training, and largely applicable, based on a inner loop in the inference phase which exploits mechanism from reinforcement learning (usually employed in alignment) and employs a score given by the model itself.
- The paper presents several experiments, among them testing their method on harmless generation, showing its efficiency through automatic and human evaluation.

**Weaknesses:**

- The paper is a little difficult to follow, and I believe section 3.1 and parts of section 3.2 could be improved for a better presentation of the method.
- While the main experiment (being the main motivation of the paper) shows convincing results, I found several issues with the others:
    - While I can see its interest, I believe the robustness experiment should be better introduced and motivated in the paper.
    - Similarly, truthfulness can obviously be linked to harmlessness - still, I would have liked this to be discussed to better motivate the experiment.
    - Lastly, I do not completely understand the relevance of controlled generation.
    - While I understand that this may already common practice, having almost all of the production of your model evaluated by pre-existing model - which, furthermore, are not even public (nor is their training data) is something I believe to be an issue and should be **at least discussed** in the paper.
- The method completely relies on a self-evaluation measure, in a form of a prompt. This dimension of the method, which to me seems like is the most capital, stays completely uninvestigated. Have other prompts be tested ? With a different keyword (harmful/harmless) ? How do naïve baselines (using a pre-trained binary classifier in place of self-evaluation, or using the current prompt without the inner loop but in the context of naive generate and evaluate) compare in term of running time and performance ?

**Questions:**

- I believe Figure 2 should be capital to the understanding of your method - however, the text helped me understanding the figure. I believe you should try to simplify what you display for the first contact of the reader with your method.
- A large part of your citations are preprints - if those papers have been published, please update their reference.
- While I appreciate your ablation studies, I believe those aspects of your method are all well-motivated. If you did not implement ablation studies for other mechanisms in your method, did you have any preliminary results showing the importance of them, or motivating your choices ?

---

> ### Author Response · Authors · 2023-11-18
>
> Thank you for your detailed review and valuable feedback, which will greatly assist us in improving our paper. Below are our responses to your queries and clarifications regarding some of the weaknesses you have pointed out.
>
> > Q1. How do naïve baselines (using a pre-trained binary classifier in place of self-evaluation, or using the current prompt without the inner loop but in the context of naive generate and evaluate) compare in term of running time and performance?
>
> We have compared RAIN with the naïve baseline, labeled as "Vanilla (Repeated)", in Table 5 on page 9 of the original manuscript. In this experiment, the naïve baseline involves conducting 500 sampling iterations, followed by selecting the optimal generation through self-evaluation. For the 7B model, the naïve baseline shows minimal improvement in the model's safety. For the 30B model, RAIN significantly outperforms the naïve baseline in terms of both efficiency and performance. This comparison illustrates that RAIN is markedly superior in enhancing model performance. A binary classifier designed to evaluate whether a text is harmful must be trained using data related to harmlessness. This is akin to training the reward model in RLHF, leading to unfair comparisons (RAIN does not utilize additional data). To ensure a fair comparison, our experimental approach is in line with your suggestion, namely "using the current prompt without the inner loop in the context of naïve generate and evaluate."
>
> > Q2. While I understand that this may already common practice, having almost all of the production of your model evaluated by pre-existing model - which, furthermore, are not even public (nor is their training data) is something I believe to be an issue and should be at least discussed in the paper.
>
> To address this potential limitation, we have not only utilized evaluations from pre-existing models but have also conducted human evaluations to validate our findings. The results of these human evaluations are presented in Table 8 on page 9 of our original manuscript. These results demonstrate a close alignment between evaluations conducted by GPT-4 and those by human assessors, indicating consistency in the assessment quality. Additionally, recent research [1] supports the reliability of GPT-4 evaluations, further validating our approach. In our experiments conducted on AdvBench [2], we used the same metrics as those in the original paper that introduced this benchmark. These metrics are not determined by model judgment but are based on explicit, predefined rules.
>
> > Q3. The method completely relies on a self-evaluation measure, in a form of a prompt. This dimension of the method, which to me seems like is the most capital, stays completely uninvestigated. Have other prompts been tested? With a different keyword (harmful/harmless)?
>
> Thank you for your suggestion. We acknowledge the importance of this aspect and agree that it could enrich our ablation study. Our original prompt already incorporates both "harmful" and "harmless" as key terms, as demonstrated in the prompt example on page 5 of our paper. To further investigate the sensitivity to variations in prompts, we conducted an additional experiment using the following alternative prompt: "This is a conversation between a human and an AI assistant. The assistant's response should comply with ethical guidelines, comply with the law, and avoid providing information that could cause harm or adverse effects. Do you think the assistant's response is harmful?" The results of this experiment are as follows:
>
> | Prompt                  | Original | Modified |
> |-------------------------|----------|----------|
> | Attack Success Rate↓    | 19%      | 18%      |
>
> Our experimental findings indicate that the utilization of different prompts with similar thematic content does not significantly alter the results, suggesting that the system’s performance is robust to prompt variations.
>
> References:
>
> [1] Pan, Alexander, et al. "Do the Rewards Justify the Means? Measuring Trade-Offs Between Rewards and Ethical Behavior in the Machiavelli Benchmark." International Conference on Machine Learning. PMLR, 2023.
>
> [2] Zou, Andy, et al. "Universal and Transferable Adversarial Attacks on Aligned Language Models." arXiv preprint arXiv:2307.15043 (2023).

---

> ### Author Response · Authors · 2023-11-18
> **Response continued**
>
> > Q4. While I can see its interest, I believe the robustness experiment should be better introduced and motivated in the paper.
>
> The significance of robustness in LLM alignment is highlighted by a study [1] introducing the Greedy Coordinate Gradient (GCG). This study reveals that even well-aligned models, including ChatGPT, are vulnerable to harmful responses under GCG attacks, underscoring the importance of ensuring safety against adversarial attacks.
>
> > Q5. Similarly, truthfulness can obviously be linked to harmlessness - still, I would have liked this to be discussed to better motivate the experiment.
>
> LLMs sometimes generate responses that are not truthful, which can potentially mislead users and lead to serious consequences. In fields like healthcare, accuracy in factual information is extremely sensitive. When applying LLMs to these areas, special attention must be paid to the truthfulness of their outputs. Truthfulness is also a current focus in LLM alignment research, with studies such as LLaMA2-chat [2] and InstructGPT [3] both investigating and testing for truthfulness.
>
> > Q6. Lastly, I do not completely understand the relevance of controlled generation.
>
> While not directly related to safety, exploring controlled sentiment generation allows us to test RAIN's versatility across a wider range of tasks. Recent works in LLM alignment, such as DPO [4] and RAFT [5], have also experimented in this task, indicating its relevance to the field.
>
> > Q7. A large part of your citations are preprints - if those papers have been published, please update their reference.
>
> Thank you for your valuable suggestion. We have thoroughly reviewed all our references and identified four papers that have since been published. These publications are:
>
> - Cho, Woon Sang, et al. "Towards Coherent and Cohesive Long-form Text Generation." Proceedings of the First Workshop on Narrative Understanding. 2019.
>
> - Kreutzer, Julia, et al. "Can Neural Machine Translation be Improved with User Feedback?." Proceedings of NAACL-HLT. 2018.
>
> - Lin, Stephanie, Jacob Hilton, and Owain Evans. "TruthfulQA: Measuring How Models Mimic Human Falsehoods." Proceedings of the 60th Annual Meeting of the Association for Computational Linguistics (Volume 1: Long Papers). 2022.
>
> - Reimers, Nils, and Iryna Gurevych. "Sentence-BERT: Sentence Embeddings using Siamese BERT-Networks." Proceedings of the 2019 Conference on Empirical Methods in Natural Language Processing and the 9th International Joint Conference on Natural Language Processing (EMNLP-IJCNLP). 2019.
>
> We are committed to updating these citations in the forthcoming version of our manuscript.
>
> > Q8. I believe Figure 2 should be capital to the understanding of your method - however, the text helped me understanding the figure. I believe you should try to simplify what you display for the first contact of the reader with your method.
>
> Thank you for your valuable feedback. We provided a simplified diagram, which abstracts away the internal details and utilizes a concrete example for clarity, in the “Supplementary Material” section of our OpenReview submission, within the .zip file, under the file name "Simplified\_Diagram.pdf". The caption for this diagram is as follows: "A simplified explanation of RAIN. RAIN switches between a forward generation phase and a backward rewinding phase, incorporating a self-evaluation stage in between to accomplish self-alignment. The method mirrors human behavioral patterns: contemplating, weighing, and reflecting on the consequences before speaking. Notably, the method operates without the need for extra data and abstains from model updates."
>
> References:
>
> [1] Zou, Andy, et al. "Universal and Transferable Adversarial Attacks on Aligned Language Models." arXiv preprint arXiv:2307.15043 (2023).
>
> [2] Touvron, Hugo, et al. "Llama 2: Open Foundation and Fine-Tuned Chat Models." arXiv preprint arXiv:2307.09288 (2023).
>
> [3] Ouyang, Long, et al. "Training language models to follow instructions with human feedback." Advances in Neural Information Processing Systems 35 (2022): 27730-27744.
>
> [4] Rafailov, Rafael, et al. "Direct Preference Optimization: Your Language Model is Secretly a Reward Model." arXiv preprint arXiv:2305.18290 (2023).
>
> [5] Dong, Hanze, et al. "RAFT: Reward rAnked FineTuning for Generative Foundation Model Alignment." arXiv preprint arXiv:2304.06767 (2023).

---

> > ### Author Response · Authors · 2023-11-18
> > **Response continued**
> >
> > > Q9. While I appreciate your ablation studies, I believe those aspects of your method are all well-motivated. If you did not implement ablation studies for other mechanisms in your method, did you have any preliminary results showing the importance of them, or motivating your choices?
> >
> > Indeed, other mechanisms are intricately interconnected, making them infeasible to isolate for individual ablation studies. For instance, the "forward step" relies on the attributes updated in the "evaluation and attribute update step." Removing or altering the "evaluation and attribute update step" would render the "forward step" completely ineffective, as it directly relies on the updates from the previous step. The motivation behind other mechanisms in RAIN is to enhance search efficiency by heuristically exploring areas more likely to lead to optimal generation, utilizing scores from simulated generations.

---

### Author Response · Authors · 2023-11-18
**General Response**

We thank all reviewers for meticulous reviews and insightful suggestions! Regarding comparisons with baselines, we highlight that our original manuscript has included detailed results of these comparisons as suggested. In particular, reviewer hgqg suggested a comparison with naïve baselines, and reviewer RrQ5 mentioned a comparison with the best-of-K method, both of which have been detailed in Table 5 on page 9 of our original manuscript. The naïve baseline and the best-of-K method are referred to as "Vanilla (Repeated)" in the table. Reviewer drDq mentioned a comparison with RLHF, which has been shown in Figure 4 on page 8 of the original manuscript.

In Table 5 on page 9, we specifically compared RAIN with the "Vanilla (Repeated)" approach, involving vanilla auto-regressive sampling conducted 500 times, followed by the selection of the optimal generation using the same self-evaluation method as RAIN. For the 7B model, "Vanilla (Repeated)" shows minimal improvement in the model’s safety. For the 30B model, RAIN demonstrates significantly better performance.

Furthermore, in Table 4 on page 8 of the original manuscript, we presented the comparison results with RLHF/RLAIF (including SFT and RL phases). RLHF necessitates human annotations, and RLAIF requires data in the form of prompts. RAIN, however, does not require either. The method achieves better performance than RLHF and RLAIF.

We hope the reviewers and AC could take these ignored experiments into account. We will also highlight them more explicitly in the forthcoming version. Thank you!

---

### Meta-Review · Area_Chair_Ua9T · 2023-12-02

**Metareview:**

This paper proposes RAIN, Rewindable Auto-regressive INference, to allow LLMs align themselves with human preferences at inference time. RAIN also enables self-alignment to eliminate the need of human annotation for alignment data, which is costly to collect. RAIN leverages a two-layer algorithm for decoding. In the outer loop, RAIN runs subsequence generation process enabled by the innner loop and concatenates generated subsequences as the final output. In the inner loop, RAIN generates subsequences via an iterative search process including 1) lookahead planning in a tree-search paradigm, and self-evaluating the newly added nodes/tokens and updating values for nodes. RAIN aims at simulating human weighing patterns, involving evaluation, backtracking, and the selection of optimal generation paths. RAIN is empirically showed effective on tasks like harm-free generation and truthful QA, and RAIN improves alignment of fixed LLMs like LLaMA without finetuning.

Strength:
 - The paper studies the important topic of aligning LLMs (for satety, trustworthy etc.), and proposes an interesting inference time solution to generate safe content without external annotated data or finetuning models. Proposed approach may reduce the development cost of LLMs in human annotation and finetuning etc. The inference time solution is novel and might open up new research opportunities.
 - RAIN is model agnostic and potentially applicable to various pretrained/finetuned LLMs
 - The paper conducts reasonable experiments and ablation study to support the effectiveness of RAIN.

Weakness:
 - The main weakness lies in the high computational cost (and the complexity in algorithm introduced in order to mitigate the cost). The complexity and cost might hinder the algorithm to be applied broadly. Also, to me, the algorithm delays the resource burden (data annotation, finetuning) to a later inference stage, while it's actually not clear which stage, finetuning or inference, is more expensive, considering finetuning needs to be done once at centralized computing clluster, while inference are invoked repeatedly after model deployment at edge devices which have significantly interferior computing power. Thus, I feel trading inference cost for finetuning free might be of research value, but not that practically reasonable. On the other hand, it is meaningful in research and practice to enable models to do self-alignment, for reducing or elimintating annotation dependency. I would love to see the authors apply RAIN to finetuning.

**Justification For Why Not Higher Score:**

Due to the weakness of inference cost, I think the work is of limited interest to ICLR community and I wouldn't suggest bump up the scores.

**Justification For Why Not Lower Score:**

Due to the weakness of inference cost, the problem setting is a bit inpractical to me and I wouldn't mind to further reduce score (reject) for this paper.

---

### Decision · Program_Chairs · 2024-01-16

Accept (poster)